# The CXCL1-CXCR2 Axis as a Component of Therapy Resistance, a Source of Side Effects in Cancer Treatment, and a Therapeutic Target

**DOI:** 10.3390/cancers17101674

**Published:** 2025-05-15

**Authors:** Jan Korbecki, Mateusz Bosiacki, Maciej Pilarczyk, Marcin Kot, Piotr Defort, Ireneusz Walaszek, Dariusz Chlubek, Irena Baranowska-Bosiacka

**Affiliations:** 1Department of Anatomy and Histology, Collegium Medicum, University of Zielona Góra, Zyty 28, 65-046 Zielona Góra, Poland; jan.korbecki@onet.eu; 2Department of Biochemistry and Medical Chemistry, Pomeranian Medical University in Szczecin, Powstańców Wlkp. 72, 70-111 Szczecin, Poland; mateusz.bosiacki@pum.edu.pl (M.B.); dchlubek@pum.edu.pl (D.C.); 3Neurosurgery Center University Hospital, Collegium Medicum, University of Zielona Gora, Zyty 28, 65-417 Zielona Gora, Poland; m.pilarczyk@inm.uz.zgora.pl (M.P.); m.kot@cm.uz.zgora.pl (M.K.); p.defort@cm.uz.zgora.pl (P.D.); 4Department of Nursing, Pomeranian Medical University in Szczecin, Żołnierska 48, 71-210 Szczecin, Poland; ireneusz.walaszek@pum.edu.pl

**Keywords:** CXCL1, CXCR2, side effect, resistance to radiotherapy, resistance to chemotherapy

## Abstract

This review examines the therapeutic potential of targeting CXCL1 and its receptor, CXCR2, in cancer treatment. It discusses anti-CXCL1 antibodies and CXCR2 antagonists, including AZD5069, SB225002, SCH-479833, navarixin/SCH-527123, ladarixin/DF2156A, and reparixin, as well as strategies to enhance CXCR2 expression in lymphocytes during adoptive cell therapy to improve immunotherapy outcomes. Particular attention is given to the role of CXCL1 in treatment resistance, including resistance to chemotherapy, radiotherapy, and anti-angiogenic therapy.

## 1. Introduction

CXC motif chemokine ligand 1 (CXCL1) is an α-chemokine characterized by a CXC motif at its N-terminus [1]. Previously known as growth-regulated gene-α (Gro-α) and melanoma growth-stimulatory activity (MGSA), CXCL1 primarily signals through the CXC motif chemokine receptor 2 (CXCR2). At higher concentrations, it can also activate CXCR1 [2]. Another receptor for CXCL1 is the atypical chemokine receptor 1 (ACKR1), also known as the Duffy antigen receptor for chemokines (DARC) [3]. However, binding to this receptor does not appear to induce significant signal transduction, though it may regulate CXCL1 availability [4].

CXCL1 plays a critical role in cancer progression by directly affecting tumor cells. It promotes cancer cell proliferation [5,6,7] and supports the self-renewal of cancer stem cells [8,9]. However, its effect on proliferation varies depending on the tumor type; in cholangiocarcinoma, CXCL1 has been shown to suppress cancer cell proliferation [10]. Additionally, CXCL1 exerts anti-apoptotic effects by modulating the expression of B-cell leukemia/lymphoma-2 (Bcl-2) family proteins [11,12]. It also enhances cancer cell migration [13,14,15] and contributes to immune evasion by upregulating programmed death-ligand 1 (PD-L1) expression in tumor cells [6].

Beyond its direct effects on cancer cells, CXCL1 influences the tumor microenvironment by acting on non-malignant cells. Due to the expression of CXCR2 on endothelial cells, CXCL1 promotes angiogenesis (Figure 1) [16,17,18]. Like other chemokines, it facilitates immune cell migration and contributes to the immunosuppressive tumor milieu. CXCL1 is involved in recruiting granulocytic myeloid-derived suppressor cells (G-MDSC) [19,20] and tumor-associated neutrophils (TAN) [21,22,23]. Moreover, CXCL1 influences tumor-associated macrophages (TAM) by promoting M2 polarization [24,25]. It also indirectly increases the presence of monocytic myeloid-derived suppressor cells (M-MDSC) in the tumor niche by stimulating their expansion in the bone marrow [26].

Although the role of CXCL1 in cancer progression is well documented, its impact on cancer therapy has received relatively little attention. This review aims to address this gap and highlight the potential of CXCL1 as a therapeutic target in combination with standard cancer treatments. The discussion focuses on CXCL1-mediated therapy resistance and its contribution to the adverse effects of cancer treatment.

## 2. Anticancer Therapy vs. CXCL1

### 2.1. CXCL1-CXCR2 Axis as a Therapeutic Target in Cancer Therapy

CXCL1 is a significant factor in cancer processes, inducing an autocrine stimulation of tumor cell proliferation [27,28] and migration [8,28]. CXCL1 chemokine induces angiogenesis [17,18,29,30] and recruitment of G-MDSCs [31,32] and neutrophils [33] into the tumor niche. Also, the expression of this chemokine is elevated relative to healthy tissues in many types of cancer [34,35,36,37,38,39,40,41,42,43]. For this reason, the CXCL1-CXCR2 axis represents a convenient therapeutic target for cancer treatment.

Many therapeutic approaches targeting the CXCL1-CXCR2 axis are currently under investigation. An example of this is HL2401, a monoclonal antibody anti-human CXCL1 [44] which inhibits the proliferation and migration of bladder and prostate cancer cells and inhibits the angiogenesis of the tested tumors. To date, no clinical trials have been conducted using HL2401.

However, a more effective therapeutic approach is the use of CXCR2 inhibitors. In addition to disrupting the CXCL1 function, this approach also targets all CXCR2 ligands. The best-known CXCR2 antagonists include SB225002: (N-(2-hydroxy-4-nitrophenyl)-N’-(2-bromophenyl)urea) [45], which has shown activity against glioma [46], androgen-independent prostate cancer [47], cervical cancer [7], chronic myeloid leukemia (CML) [48], nasopharyngeal carcinoma [49], oral squamous cell carcinoma [50], triple negative breast cancer [51], ovarian cancer [52], and cholangiocellular carcinoma [53]. At the same time, SB225002 not only decreases CXCR2 activity but also binds to β-tubulin [54,55], which leads to the disruption of microtubules, which may account in part for the anticancer properties of this compound. To date, no clinical trials have been conducted using SB225002.

Another CXCR2 inhibitor is AZD5069, a reversible antagonist of CXCR2 [56]. In vitro studies have demonstrated its antitumor activity against thyroid cancer cells [57]. In models of triple-negative breast cancer, it reverses doxorubicin resistance [58], likely by reducing the production of transforming growth factor (TGF)-β by cancer cells. AZD5069 also enhances the efficacy of anti-PD-L1 antibodies in an in vivo model of hepatocellular carcinoma [59]. AZD5069 is currently undergoing clinical evaluation [60] and has been shown to be well tolerated by patients [61]. It is being investigated for the treatment of asthma (ClinicalTrials.gov ID: NCT01704495) [62]; however, clinical trials have not demonstrated efficacy in this indication. It has also been tested in patients with chronic obstructive pulmonary disease (COPD) (ClinicalTrials.gov ID: NCT01233232) [61] and bronchiectasis (ClinicalTrials.gov ID: NCT01255592) [63]. While the compound was well tolerated, it did not lead to significant clinical improvement.

AZD5069 is also being evaluated as a potential anticancer agent. It is currently undergoing testing in combination with the anti-PD-L1 antibody durvalumab in patients with metastatic pancreatic ductal adenocarcinoma (PDAC) (ClinicalTrials.gov ID: NCT02583477), although the results of this trial have not yet been published. A similar ongoing study (ClinicalTrials.gov ID: NCT02499328) is assessing AZD5069 alone or in combination with durvalumab in patients with advanced head and neck squamous cell carcinoma (HNSCC).

Also being tested are dual antagonists for CXCR1 and CXCR2. These compounds inhibit the activity of a receptor for CXCL1 and also inhibit the activity of CXCR1, which means that they also reduce the activity of CXCL8/IL-8. Such a compound is SCH-479833 [64,65]. SCH-479833 has been shown to have anticancer properties for melanoma [64] and colon cancer [65]. However, this compound has not yet undergone clinical testing.

Another dual antagonist of CXCR1 and CXCR2 is navarixin (SCH-527123, MK-7123) [66,67], which also inhibits CCR7 [68]. Navarixin has demonstrated antitumor activity against several cancer types, including colorectal cancer [65,69] and melanoma [64,70]. It is currently being investigated in clinical trials as a potential treatment for COPD (ClinicalTrials.gov ID: NCT01068145, NCT00441701) and asthma (ClinicalTrials.gov ID: NCT01006161, NCT00632502), as well as for psoriasis (ClinicalTrials.gov ID: NCT00684593). Navarixin has also been evaluated in oncology (ClinicalTrials.gov ID: NCT03473925) [71]. In this trial, navarixin was administered in combination with pembrolizumab to 105 patients with various cancer types. Pembrolizumab is an anti-PD-1 antibody, and the treatment aimed to simultaneously block CXCR1/CXCR2 signaling and a key immune checkpoint involved in tumor immune evasion. However, the combination failed to produce a satisfactory therapeutic response. The most severe adverse events reported included neutropenia, hepatitis, and pneumonitis [71]. These results suggest that CXCR2 inhibitors, whether used alone or in combination with PD-1-targeted therapies, may lack sufficient therapeutic efficacy. One possible explanation is the off-target inhibition of CCR7 by navarixin [68]. CCR7 plays a critical role in directing lymphocyte trafficking to lymph nodes [72]; thus, its disruption may impair immune system function and counteract the effects of PD-1 blockade.

Another compound being tested is ladarixin/DF2156A [73,74,75], a non-competitive allosteric inhibitor of CXCR1 and CXCR2 receptors, which has demonstrated anti-tumor properties for melanoma [75] and PDAC [76]. Ladarixin is being evaluated in clinical trials as a potential treatment for type 1 diabetes (ClinicalTrials.gov IDs: NCT05368402, NCT02814838, NCT04899271, NCT04628481, NCT05035368). In these studies, it did not significantly prevent β-cell loss in most patients [77], although further investigations into its use for type 1 diabetes are ongoing [78]. Ladarixin is also being tested in combination with sotorasib—a targeted inhibitor of mutant K-RAS—in patients with KRAS G12C-mutant non-small-cell lung cancer (NSCLC) (ClinicalTrials.gov ID: NCT05815173). This trial is currently in the patient recruitment phase.

Another dual CXCR1/CXCR2 antagonist is reparixin (Figure 2), a non-competitive allosteric inhibitor that preferentially binds and stabilizes the inactive conformation of both receptors [79,80]. In vitro studies have shown that reparixin exhibits antitumor activity in thyroid cancer [81] and pancreatic cancer [82]. It has also demonstrated both in vitro and in vivo efficacy against breast cancer [83], at least in part by reducing the self-renewal capacity of cancer stem cells (CSCs), thereby enhancing the therapeutic effect of docetaxel. Reparixin is currently undergoing clinical evaluation as a potential anticancer agent. In a Phase I trial in patients with HER2-negative breast cancer (ClinicalTrials.gov ID: NCT01861054), the compound was well tolerated and exhibited minimal toxicity [84]. In another trial, reparixin was tested in combination with paclitaxel in 30 patients with HER2-negative metastatic breast cancer (ClinicalTrials.gov ID: NCT02001974) [85], where it maintained a favorable safety profile and was associated with a 30% response rate. However, in a separate clinical study involving patients with metastatic triple-negative breast cancer (ClinicalTrials.gov ID: NCT02370238), reparixin did not enhance the therapeutic efficacy of paclitaxel [86], though it continued to show low toxicity.

The antibody LY3041658 has been developed to neutralize all known ligands of the CXCR2 receptor [87]. Rather than acting directly on the receptor, it reduces CXCR2 activation by inactivating its ligands. LY3041658 is currently undergoing early-stage clinical trials in healthy volunteers (ClinicalTrials.gov ID: NCT02148627, NCT04653168).

In addition to reducing the activation of the CXCL1-CXCR2 axis in tumors, this axis can be used as part of anti-cancer therapy using the conjugation of daunorubicin with CXCL1 [63], where CXCL1 serves as a transporter for daunorubicin (Table 1). Cancer cells have high expression of CXCR2, so CXCL1 will bind with high affinity to its receptor on the membrane of cancer cells. Along with this chemokine, the cancer cell will take up the conjugated drug. This approach has been investigated in an in vitro melanoma model [88].

Inhibition of CXCR2 abolishes all tumor processes involving the CXCL1-CXCR2 axis. However, it is possible to transform pro-cancerous mechanisms into anti-cancerous ones. An example of this is the polarization of neutrophils [89], cells which have either N1 or N2 polarity depending on the factors to which they are exposed. Neutrophils with N2 polarity are pro-tumor cells that contribute to cancerous tumor growth. In contrast, neutrophils with N1 polarity are anti-cancer cells. By using pro-inflammatory compounds, it is possible to recruit neutrophils through the CXCL1-CXCR2 axis and then polarize these cells into anti-tumor neutrophils with N1 polarity. OM-174, which is a lipid A analog, can be used for this [90]; alternatively, such therapy can be combined with a standard chemotherapeutic agent, such as oxaliplatin or cisplatin.

None of the compounds targeting CXCL1 or CXCR2 listed above have been approved by the US Food and Drug Administration (FDA) as official anticancer drugs. However, they are being evaluated in in vitro and in vivo studies as well as in early-phase clinical trials.

### 2.2. Herbal Substances with Anticancer Activity Against CXCL1

Some anticancer drugs and therapeutic substances act against CXCL1. An example of this is curcumin, which decreases nuclear factor κB (NF-κB) activation and thus reduces the expression of genes dependent on this transcription factor, including *CXCL1* [91]. Curcumin also increases the expression of miR-181b, which directly downregulates CXCL1 mRNA [92]. This effect also occurs when curcumin is combined with 5-fluorouracil, a drug in which some side effects are due to an increase in CXCL1 expression induced by this drug. The use of curcumin abolishes this increase and improves the effectiveness of 5-fluorouracil in cancer treatment [93].

Another compound of natural origin is ginsenoside panaxatriol, one of the main active ingredients of Asian ginseng (*Panax ginseng*) preparations [94]. Ginsenoside panaxatriol reduces the activation of NF-κB in breast cancer cells and thus decreases the expression of chemokines, such as CXCL1 and CXC motif chemokine ligand 8 (CXCL8)/interleukin-8 (IL-8), increasing the effectiveness of paclitaxel [94].

The XIAOPI formula consists of 10 herbs and 196 compounds [95]. One of the compounds identified in this drug is baohuoside I, which downregulates CXCL1 expression in TAM [96,97]. For this reason, the XIAOPI formula acts against breast cancer. In particular, the XIAOPI formula interferes with the function of cancer stem cells [98] and decreases autophagy in cancer cells [99], which increases chemosensitivity during the use of anticancer drugs such as paclitaxel.

### 2.3. Adoptive Cell Therapy

Adoptive cell therapy is a type of immunotherapy which involves taking lymphocytes from a patient, modifying them to fight the cancer more effectively, and then introducing them back into the patient’s body [100]. In the first protocols, lymphocytes were isolated from the cancerous tumor, that is, lymphocytes that had the ability to infiltrate the cancerous tumor and destroy cancer cells. These lymphocytes were then multiplied and introduced back into the patient’s body. Currently, researchers are testing lymphocytes with either modified T cell receptors (TCR) or expressed chimeric antigen receptors (CARs) recognizing specific cancer antigens.

The modified lymphocytes recognize tumor antigens and thus effectively destroy tumor cells. However, infiltration of lymphocytes into a solid tumor is hampered because such lymphocytes have low-level expression of receptors responsible for their migration into the tumor niche. An example of such a receptor is CXCR2 [101,102,103,104]—the most important receptor for CXCL1. Expression of CXCL1 is upregulated relative to healthy tissue in many cancerous tumors such as bladder cancer [39], colon cancer [38,42], gastric cancer [36], hepatocellular carcinoma [35], non-small-cell lung carcinoma [40], melanoma [34], ovarian cancer [37], PDAC [43], and renal cell carcinoma [105]. In breast cancer, CXCL1 expression is reduced relative to healthy tissue [41]. Conversely, it is elevated in triple-negative breast cancer [106].

Due to increased CXCL1 expression in many cancers, CXCR2 transduction into lymphocytes increases tumor infiltration by such modified cells, which was confirmed by experiments on T cells with increased CXCR2 expression, which specifically migrated to ovarian cells [102]. These modified T cells exhibit improved infiltration into tumors in in vivo models of melanoma [103,107] and colon cancer [107]. At the same time, CXCR2 activation on such lymphocytes caused an increase in interferon-γ (IFN-γ) secretion [101], a cytokine that enhances the anti-tumor immune response.

CXCR2 transduction, as an additional modification of lymphocytes, improves the effectiveness of therapy. An example of this is the transduction of CXCR2 into either CAR-T cells or T cells with modified TCR. Such additional modification induces such modified T cells to specifically accumulate in solid tumors, as confirmed by experiments on ovarian cancer [108], hepatocellular carcinoma [104], glioblastoma, and PDAC [108]. This efficacy can also be improved by the use of radiation therapy [108]. Ionizing radiation increases CXCL1 expression in many cancers, such as breast cancer [109], glioblastoma multiforme [108,110,111,112,113], and non-small-cell lung cancer [114]. This may increase the infiltration of T cells with increased CXCR2 expression.

In addition to T cells, natural killer (NK) cells are also being tested as an example of adoptive cell therapy (Table 2). Increasing CXCR2 expression on these cells causes tumor infiltration by these cells, giving an anti-tumor effect, as shown in experiments in vitro on renal cell carcinoma [115].

On the other hand, it should be mentioned that CXCL1 causes the recruitment of G-MDSCs into the tumor niche [116,117,118]—cells that inhibit the action of anti-tumor lymphocytes and thus reduce the effectiveness of immunotherapy. For this reason, to increase the effectiveness of adoptive cell therapy, one should use modified lymphocytes with drugs that target cancer immune evasion.

### 2.4. Photodynamic Therapy

Photodynamic therapy involves using a photosensitizer that accumulates in the tumor and then applying light that reacts with the compound [119]. This results in the production of ROS that damage the cancer cells. Photodynamic therapy also causes inflammation, which activates the immune system to destroy cancer cells. This is because fibroblasts secrete heat shock protein family A member 1B (Hspa1b) in response to photodynamic therapy [120]. Hspa1b, dependent on toll-like receptors (TLR)2 and TLR4, activates NF-κB in macrophages, which increases the expression of pro-inflammatory cytokines such as interleukin-1β (IL-1β) and tumor necrosis factor-α (TNF-α), as well as CXCR2 ligands in these cells, as shown in experiments on mouse models.

Photodynamic therapy also increases the expression of CXCR2 ligands in tumorigenesis, as shown in experiments on breast cancer in mice [121]. In mice with murine colon carcinoma, it induces the accumulation of T helper type 17 (Th17) cells in tumor-draining lymph nodes [122], which increases the expression of macrophage inflammatory protein-2 (MIP-2) (but not keratinocyte-derived chemokine (KC)) that causes the infiltration of neutrophils into tumor-draining lymph nodes. Neutrophils in this location enhance the anti-tumor immune response.

In mouse models, MIP-2 (but not KC) plays an important role in neutrophil recruitment [121,122]. Studies in human models are lacking, and for this reason it is not known which of the CXCR2 ligands in humans have a significant role in the induction of the anti-tumor immune response.

## 3. Resistance to Therapy

### 3.1. Resistance to Chemotherapy

CXCL1 expression is increased by chemotherapeutics such as the histone deacetylase (HDAC) inhibitor Belinostat in breast cancer cells [123], paclitaxel in triple-negative breast cancer cells [94] and in melanoma cells [124], 5-fluorouracil in murine 4T1 triple-negative breast cancer cells [125], oxaliplatin in colorectal cancer cells [126] and metastatic castration-resistant prostate cancer cells [127], epidoxorubicin in bladder cancer cells [128], doxorubicin in triple-negative breast cancer cells [129], and carboplatin in murine B16-F10 melanoma cells [130]. Also, chemotherapeutics increase the expression of other CXCR2 ligands, including CXCL8/IL-8 [127]. However, not every anticancer drug induces the CXCL1-CXCR2 axis activation, and also not in every type of cancer. For example, taxanes cause a decrease in CXCR2 expression in metastatic castration-resistant prostate cancer [131], which reduces the action of CXCL1.

An increase in CXCL1 expression by chemotherapeutics is associated with NF-κB activation (Figure 3) [126,129], although this effect may depend on an increase in TNF-α production [132] and, at least in the case of doxorubicin, the activation of NF-κB depends on p53 deficiency [129]. Increased expression of CXCL1 as well as other CXCR2 ligands results in autocrine increases in the expression of these chemokines [133].

CXCR2 activation is responsible for resistance to chemotherapy. In particular, it causes increased activation of NF-κB, which leads to the increased expression of Bcl-2 [126,127,131] and baculoviral IAP repeat containing 5 (BIRC5)/survivin [127]. NF-κB can also directly increase the expression of other anti-apoptotic Bcl-2 family proteins, for example, Bcl-2-related gene expressed in fetal liver-1 (Bfl-1)/A1 [134] and B-cell lymphoma-extra-large (Bcl-x_L_) [135]. Therefore, this may be the main mechanism of chemoresistance. Also, CXCR2 ligand-induced chemoresistance may be mediated by neutrophils and G-MDSCs [132,136]. G-MDSCs are recruited to the tumor niche by the CXCR2 ligands. CXCL1 present in extracellular vesicles derived from apoptotic cancer cells also contributes to tumor immune evasion [137]. It enhances PD-1 activity and promotes M2 polarization in TAMs. In addition, CXCL1 increases PD-L1 expression on cancer cells, thereby suppressing the activity of cytotoxic lymphocytes [138]. G-MDSCs and M2 TAM may inhibit the anti-cancer response of the immune system induced by the use of chemotherapeutics.

Another drug resistance mechanism involving CXCL1 is autophagy-mediated chemoresistance [139]. This process has been observed in experimental models of breast cancer and is specifically linked to resistance to paclitaxel.

Due to the pro-survival effects of CXCR2 ligands, it is possible to use a standard chemotherapeutic agent together with a CXCR2 inhibitor to increase the efficacy of the applied therapy. Therefore, blocking CXCR2 activity or inactivating CXCR2 ligands increases the efficacy of drugs such as 5-fluorouracil [140], paclitaxel [133,141], doxorubicin [133,141], oxaliplatin [69,126], and cisplatin [131,136] on in vitro models and in experimental animal tests.

Importantly, the increase in the expression of CXCL1 and other CXCR2 ligands may not be a mechanism of chemoresistance but only a marker of chemoresistance in some cancers associated with the induction of TNF-α expression by chemotherapeutics, e.g., docetaxel [142]. There may be two types of receptors for TNF-α on a tumor cell: tumor necrosis factor receptor (TNFR)1 and TNFR2. TNFR1 has pro-apoptotic properties, while TNFR2 exerts a pro-survival effect, causing the activation of NF-κB and increasing the expression of CXCL1. This means that an increase in CXCL1 expression can be a marker of TNFR2 expression on a tumor cell and decreased TNFR1 expression. In this way, CXCL1 is a marker of the lack of pro-apoptotic effect of TNF-α, whose expression is induced by chemotherapeutics [142].

### 3.2. Resistance to Radiotherapy

Radiation therapy is currently one of the most significant therapeutic approaches in cancer treatment [143]. It involves the use of ionizing radiation to damage cellular DNA and inhibit the division of rapidly dividing cancer cells, resulting in their destruction. Nevertheless, cancers have developed many radioresistant mechanisms. One such example is the cancer stem cells, which divide infrequently and exhibit an enhanced ability to repair DNA and protect against reactive oxygen species (ROS). Other mechanisms of radioresistance involve growth factors in the tumor microenvironment that have a pro-survival effect, for example, CXCL1, which activates extracellular signal-regulated kinase (ERK) mitogen-activated protein kinase (MAPK) and thus has a pro-survival effect on cancer cells exposed to ionizing radiation [144]. CXCL1 also decreases the expression of superoxide dismutase 1 (SOD1) in cancer cells, as shown by experiments on esophageal squamous cell carcinoma cells, which results in increased levels of ROS in the tumor cell and thus increased activity of DNA damage repair enzymes [144]. Finally, CXCL1 induces the recruitment of neutrophils, which can induce radioresistance, into the tumor niche [145].

CXCL1 expression is at a high level in the tumors of many cancers [34,35,36,37,38,39,40,41,42,43]. An increase in CXCL1 expression in the tumor is also observed under the influence of ionizing radiation, mainly due to an increase in the expression of this chemokine and CXCR2 in cancer-associated fibroblasts (CAF), as shown by experiments on esophageal squamous cell carcinoma [144]. At the same time, the increase in CXCL1 expression may be the result of ionizing radiation. The fact that 1 to 5 centigrays (cGy) increase CXCL1 expression in normal human HFLIII fibroblasts [146] indicates that ionizing radiation may lead to radioresistance. Also, ionizing radiation increases CXCL1 expression in breast cancer cells [109], glioblastoma multiforme cells [108,110,111,112,113], and non-small-cell lung cancer cells [114], resulting in a decreased susceptibility of tumor cells to ionizing radiation. Studies on glioblastoma multiforme cells have shown that the increase in CXCL1 expression by ionizing radiation is casein kinase 1 alpha 1 (CK1α) dependent [113] and dependent on an increase in inhibitor of NF-κBζ (IκBζ) expression [110]. The increase in CXCL1 expression following ionizing radiation can persist for up to 35 days [111]. These effects of ionizing radiation on cancer cells may also be lineage dependent. A study on squamous cell carcinoma of head and neck cells showed that only 1 out of the 16 tested lines showed increased CXCL1 expression under ionizing radiation, and 3 of the 16 tested lines showed decreased expression of this chemokine [147].

Ionizing radiation increases the expression of other CXCR2 ligands, including CXCL2 and CXCL8/IL-8 [108]. The expression of other ligands may be higher than CXCL1. In glioblastoma multiforme cells, there was a much greater increase in CXCL8/IL-8 expression compared to CXCL1 [108].

The increase in CXCL1 expression following radiotherapy induces radioresistance and thus reduces the efficacy of the applied treatment. Also, CXCL1 causes the migration of tumor cells. For this reason, radiotherapy can also induce metastasis and cancer tumor growth, as confirmed by studies on non-small-cell lung cancer A549 cells [114]. Studies on human umbilical vein endothelial cells (HUVEC) have shown that ionizing radiation increases CXCL1 expression in these cells [148], which indicates that radiation therapy can induce tumor angiogenesis and thus tumor growth.

Blood CXCL1 levels after radiotherapy can be used as a diagnostic tool. For example, patients with hepatocellular carcinoma who had low blood CXCL1 levels after stereotactic body radiotherapy were more likely to have liver toxicity 3 months following the therapy [149].

### 3.3. Resistance to Anti-Vascular Endothelial Growth Factor Therapy

Cancer treatment also involves the use of drugs targeting angiogenesis. The earliest drug developed in anti-angiogenic therapy was bevacizumab, a humanized monoclonal antibody that inactivates vascular endothelial growth factor (VEGF)-A [150]. Currently, this drug has been approved for the treatment of many types of cancer. However, the tumor microenvironment contains several other pro-angiogenic factors which can take over the function of VEGF when it is missing.

These alternative pro-angiogenic factors include CXCL1 and other CXCR2 ligands [17,18,29,30]. CXCL1 also induces the recruitment of endothelial progenitor cells (EPC) into the tumor niche, as shown in experiments on glioblastoma multiforme [151,152,153]. CXCL1 induces the proliferation and formation of new blood vessels by recruiting EPC. In addition, glioblastoma multiforme tumors contain CXCR2^+^ cancer stem cells [154], cells that exhibit vascular mimicry due to their ability to incorporate into forming blood vessels. For this reason, high CXCL1 expression in the tumor is a factor that enhances resistance to anti-angiogenic therapies. The expression of CXCR2 ligands can also be increased by anti-angiogenic drugs, which abrogates the effect of the anti-angiogenic therapy. An example of this is vatalanib, a tyrosine kinase inhibitor of vascular endothelial growth factor receptor (VEGFR) used in experiments on mice with glioblastoma multiforme tumors [154].

The aforementioned properties of CXCR2 ligands are independent of VEGF and are the source of resistance to anti-angiogenic therapies [153]. For this reason, the use of CXCR2 inhibitors in combination with a drug targeting VEGF yields better results, as shown by experiments on glioblastoma multiforme [155] and ovarian cancer [156].

### 3.4. Resistance to Immunotherapy

CXCR2 ligands, including CXCL1, play a role in tumor immune evasion. This effect is partly mediated by the recruitment of G-MDSCs to the tumor microenvironment [21,22,23,157], where they exert immunosuppressive functions. CXCL1 also promotes M2 polarization of TAMs [24,25], a phenotype associated with immune suppression. In addition, CXCR2 ligands such as CXCL1 can induce autophagy-driven degradation of MHC class I molecules, as observed in colorectal cancer models [158], reducing the ability of immune cells to recognize and eliminate tumor cells. CXCL1 further contributes to immune evasion by upregulating PD-L1 expression on cancer cells, thereby inhibiting cytotoxic lymphocyte activity [138].

These mechanisms collectively contribute to resistance to immunotherapy. For example, elevated blood levels of CXCL8 in patients with esophageal cancer have been associated with poor outcomes following immunotherapy [159]. Similar findings were reported in biliary tract cancers, where high serum concentrations of CXCL1 and CXCL5 correlated with treatment failure in patients receiving camrelizumab (an anti-PD-1 antibody) in combination with gemcitabine and oxaliplatin [160]. In vivo studies in animal models have shown that neutralizing antibodies targeting CXCR2 ligands can enhance the efficacy of anti-PD-L1 therapy in triple-negative breast cancer [161] and glioma [162].

Several clinical trials are exploring combination therapies that target both PD-L1 and the CXCR2 signaling axis. One such trial is evaluating AZD5069 in combination with durvalumab in patients with advanced HNSCC (ClinicalTrials.gov ID: NCT02499328). Another study tested the combination of navarixin with pembrolizumab in patients with various malignancies [71], but the results were disappointing—navarixin failed to enhance the therapeutic activity of pembrolizumab.

Tumor immune evasion involves multiple redundant pathways and mechanisms; inhibiting a single pathway may not be sufficient to reverse immune suppression. While CXCR2 ligands contribute to this process, they may not be critical for maintaining immune evasion in all tumor types.

## 4. Side Effect of Chemotherapy

### 4.1. Metastasis as a Side Effect of Chemotherapy

Drugs used in chemotherapy may not so much treat cancer but also contribute to the development of cancer by inducing the migration of cancer cells and thus metastasis. This has been found in the use of drugs such as topoisomerase inhibitors [163], 5-fluorouracil [125], epidoxorubicin [128], as well as paclitaxel and carboplatin [130]. This effect depends on the type of tumor, as 5-fluorouracil induces metastasis on a 4T1 mouse triple-negative breast cancer cell line model [125] but not on a murine B16-F10 melanoma cell model [130].

The induction of tumor cell migration by chemotherapy drugs is related in part to CXCL1 and other CXCR2 ligands. Different drugs have different mechanisms of action leading to metastasis. 5-fluorouracil activates NF-κB in murine 4T1 triple-negative breast cancer cells, resulting in the increased expression of CXCR2 ligands in the tumor [125]. The same mechanism of increased CXCL1 expression has been observed for epidoxorubicin acting on bladder cancer cells [128]. Also, paclitaxel and carboplatin increase the expression of CXCR2 ligands, as found in a murine B16-F10 melanoma cell model [130]. Anticancer drugs can increase CXCL1 expression independently of NF-κB. Topoisomerase inhibitors increase ROS levels in the cell [163], which leads to the activation of Janus tyrosine kinase 2 (JAK2)-signal transducer and activator of transcription 1 (STAT1) and thus elevated CXCL1 expression, as shown in experiments on many types of cancer [163].

CXCL1, through its receptor CXCR2, induces the activation of Snail, which leads to an epithelial-to-mesenchymal transition (EMT) and the migration of tumor cells [128]. Also, CXCL1 causes the recruitment of neutrophils into the tumor niche; these cells secrete prokineticin-2, which causes tumor cell migration and subsequent metastasis. Anticancer drugs, such as 5-fluorouracil, also increase the expression of CXCR2 ligands in the lung, which results in metastasis in this organ [125].

CXCR2 ligands are an important factor in the process where chemotherapy drugs cause the migration of cancer cells and thus metastasis. For this reason, the use of anticancer drugs in combination with a CXCR2 inhibitor, such as SB265610, prevents the formation of metastasis following treatment [125,130]. It is also important to note that other factors may contribute to therapy-induced metastasis. One example is CCL2, which recruits M-MDSCs to the lungs, promoting the formation of metastases in this organ by breast cancer cells [164].

### 4.2. Chemotherapy vs. Neuropathy

Chemotherapy induces neuropathy. It is estimated that approximately 30–68% of patients treated for cancer have chemotherapy-induced peripheral neuropathy (CIPN) as a side effect of treatment [165,166]. Drugs that induce this side effect include paclitaxel, platinum compounds, and vinca alkaloids.

CIPN is partly associated with neuroinflammation of neural tissue in the dorsal root ganglia [167]. An important component of CIPN is CXCL1. Studies in mice have shown that paclitaxel increases the levels of CXCR2 ligands in the blood [167] and in the dorsal root ganglion and spinal cord [168]. In part, the expression of CXCR2 ligands in the dorsal root ganglion is caused by the infiltration of this part of the nervous system by macrophages, which produce and secrete those ligands [167]. Infiltration by macrophages is caused by an increase in CC motif chemokine ligand 2 (CCL2)/monocyte chemoattractant protein 1 (MCP-1) expression by neurons in the dorsal root ganglion [169]. CXCR2 receptor ligands, as well as the CXCR2 receptor itself, are important components of CIPN. For this reason, the use of a CXCR2 antagonist (SB225002) negates paclitaxel-induced CIPN [167].

Chemotherapy drugs can also cause neuroinflammation in the brain, resulting in cognitive impairments in treated patients [170]. An example of such a drug is doxorubicin, which in rats caused an increase in the expression of interleukin-6 (IL-6) and CXCR2 ligands (but not TNF-α) in the brain. As a consequence, the rats showed behavioral changes, although it is not known whether this was associated with a change in the expression of CXCR2 ligands in the brain, and if so, which chemokine in the treated animals was responsible for the cognitive impairments.

### 4.3. Nephrotoxicity of Chemotherapy

Current chemotherapy has many side effects. An example of this is chemotherapy-induced peripheral nephropathy after using paclitaxel, oxaliplatin, or cisplatin. This side effect is estimated to affect 70% of patients treated with such chemotherapeutics [171,172]. In the case of cisplatin, it is related to the accumulation of this drug in the kidneys, more precisely in the proximal and distal tubules [173,174]. Cisplatin damages mitochondria, resulting in the apoptosis of kidney cells. It also causes the activation of NF-κB in kidney cells and expression of pro-inflammatory cytokines in these cells [174], which results in inflammatory reactions in the kidneys that damage these organs.

One of the components of cisplatin’s effect on the kidneys is CXCL1. Cisplatin, through the activation of p38 MAPK and NF-κB, increases CXCL1 expression [175,176]. In particular, an increase in the expression of CXCL1 occurs in endothelial or tubular epithelium cells. Also, the source of CXCL1 in the kidney may be CD4^+^ T cells [177]. An increase in CXCL1 expression leads to the infiltration of the kidney by neutrophils, which are involved in inflammatory reactions that cause chemotherapy-induced peripheral neuropathy. At the same time, the infiltration of the kidneys by neutrophils during chemotherapy with cisplatin may also be mediated by leukotriene B_4_ (LTB_4_) [178].

### 4.4. Diarrhea as a Side Effect of Chemotherapy

Another side effect of chemotherapy is diarrhea. For example, 5-fluorouracil reduces the expression of aquaporin 4 (AQP4) and aquaporin 8 (AQP8) in the colon [179], which leads to diarrhea. The exact mechanism of this side effect is dependent on CXCR2 ligands, as shown by experiments in mice [93,179,180]. These chemokines induce the recruitment of neutrophils to the colon, where these cells are an important factor in reducing the expression of the aforementioned AQP4 and AQP8.

### 4.5. Cardiotoxicity as a Side Effect of Chemotherapy

Chemotherapy can also cause cardiotoxicity, for example when using doxorubicin [181]. CXCL1 can be a marker of cardiotoxicity caused by doxorubicin, as shown in a study of breast cancer patients who were treated with this chemotherapeutic agent. Cardiotoxicity was observed in those who had reduced CXCL1 levels following the first cycle of chemotherapy with doxorubicin, relative to pre-treatment levels [181]. Nevertheless, the association between reduced CXCL1 levels and cardiotoxicity is unclear. Patients with heart failure [182] and chronic ischemic heart disease [183] have elevated levels of CXCL1 in their blood, which indicates that increased, not decreased, blood levels of CXCL1 should be a marker of cardiotoxicity.

## 5. Conclusions

CXCL1 and the broader CXCR2 axis play a key role in tumor progression. They are also implicated in many of the side effects associated with standard chemotherapy and radiotherapy, as well as in resistance to anticancer treatment. A number of compounds targeting CXCL1 or its receptor CXCR2 have already been developed, most of which exhibit low toxicity. Given this, these agents represent promising candidates for combination with conventional cancer therapies. Such an approach should be further investigated for its potential to overcome therapeutic resistance and alleviate treatment-related side effects.

## Figures and Tables

**Figure 1 cancers-17-01674-f001:**
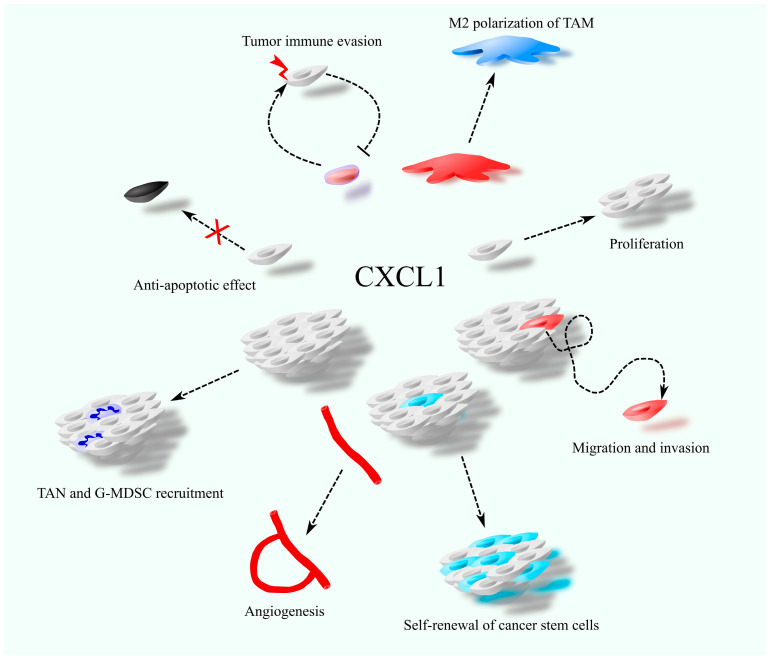
The involvement of CXCL1 in cancer-related processes, including cancer cell proliferation and migration, self-renewal of cancer stem cells, anti-apoptotic effects, angiogenesis, recruitment of tumor-associated neutrophils (TAN), polarization of tumor-associated macrophages (TAM), and tumor immune evasion.

**Figure 2 cancers-17-01674-f002:**
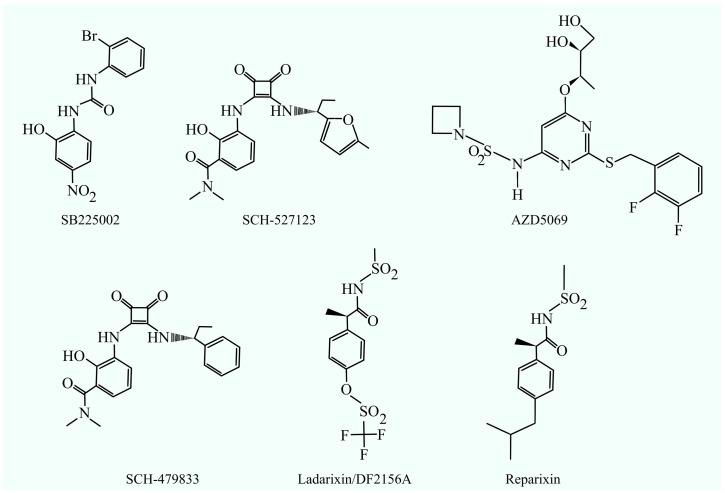
CXCR2 inhibitors: SB225002, SCH-527123, AZD5069, SCH-479833, ladarixin (DF2156A), and reparixin.

**Figure 3 cancers-17-01674-f003:**
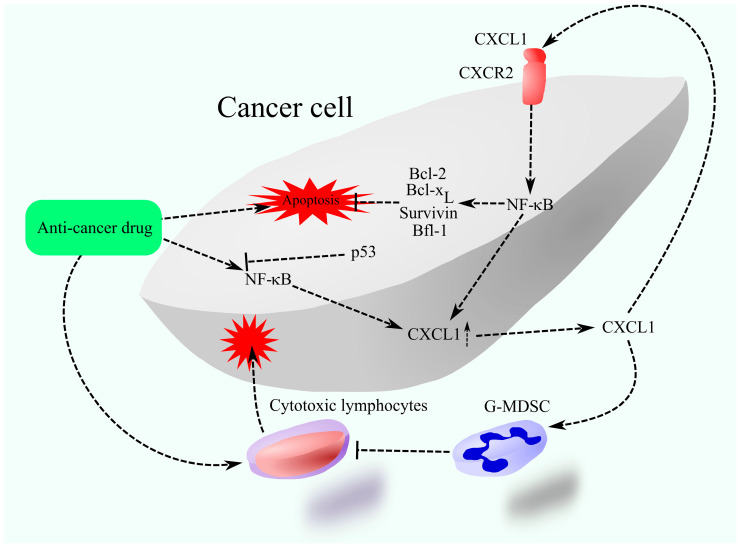
CXCL1 as a mediator of treatment resistance. Chemotherapy elevates CXCL1 expression, which, through CXCR2 activation on cancer cells, enhances the expression of anti-apoptotic proteins such as those from the Bcl-2 family and survivin, thus inhibiting therapy-induced apoptosis. Moreover, CXCL1 facilitates the recruitment of G-MDSCs into the tumor microenvironment, obstructing the ability of therapy-activated cytotoxic lymphocytes to eliminate cancer cells.

**Table 1 cancers-17-01674-t001:** Compounds targeting CXCL1 or its receptor in preclinical studies or clinical trials.

Substance	Results from In Vitro and In Vivo Studies	Results from Clinical Trials	Source
Monoclonal antibody anti-human CXCL1; HL2401	Antitumor properties against bladder and prostate cancer cells.		[44]
CXCR2 inhibitors; SB225002	Antitumor properties against glioma, androgen-independent prostate cancer, cervical cancer, chronic myeloid leukemia (CML), nasopharyngeal carcinoma, oral squamous cell carcinoma, ovarian cancer, triple-negative breast cancer, and cholangiocellular carcinoma.		[7,46,47,48,49,50,51,52,53]
CXCR2 inhibitors; AZD5069	Antitumor properties against thyroid cancer cells; reversal of doxorubicin resistance in triple-negative breast cancer.	Well tolerated in patients; evaluated in combination with the anti-PD-L1 antibody durvalumab.	[57,58,61]
Dual antagonists for CXCR1 and CXCR2; navarixin (SCH-527123, MK-7123)	Antitumor properties against melanoma and colorectal cancer.		[64,66,67,69,70,71]
Dual antagonists for CXCR1 and CXCR2; SCH-479833	Antitumor properties against melanoma and colon cancer.	Associated with high toxicity, including neutropenia, hepatitis, and pneumonitis. Shows no therapeutic effect as monotherapy or in combination with PD-L1 inhibitors.	[64,65,71]
Non-competitive allosteric inhibitor of CXCR1 and CXCR2: ladarixin/DF2156A	Antitumor properties against melanoma and PDAC.	Clinically tested against KRAS G12C-mutant NSCLC in combination with sotorasib.	[75,76]
Non-competitive allosteric inhibitor of CXCR1 and CXCR2: reparixin	Antitumor properties against breast cancer, thyroid cancer, and pancreatic cancer cells.	Exhibits low toxicity. In patients with HER2-negative metastatic breast cancer, its combination with paclitaxel was associated with a high response rate. However, in patients with metastatic triple-negative breast cancer, it did not enhance the therapeutic effect of paclitaxel.	[81,82,83,84,85,86]
Anti-CXCR2 ligand monoclonal antibody: LY3041658		Clinically tested	[87]
CXCL1 conjugated to daunorubicin	Activity against melanoma cells.		[88]

**Table 2 cancers-17-01674-t002:** CXCR2 in anticancer adoptive cell therapy.

Name	Efficacy	Mechanisms of Action	Source
Increased CXCR2 expression on T cells	Demonstrated efficacy in vitro against ovarian tumors and in vivo against melanoma and colon tumors.	T cells with increased CXCR2 expression specifically migrated toward cancer cells. This modification enhances tumor infiltration by the engineered T cells. Activation of CXCR2 on lymphocytes also increases their functional activity.	[101,102,103,107]
Increased CXCR2 expression on CAR-T cells	Demonstrated efficacy in vitro and in vivo against hepatocellular carcinoma, ovarian cancer, glioblastoma, and PDAC.	CAR-T cells with enhanced CXCR2 expression exhibit improved migration toward cancer cells and increased tumor infiltration.	[104,108]
Increased CXCR2 expression on NK cells	Demonstrated efficacy in vitro against renal cell carcinoma cells.	Migration of modified NK cells toward cancer cells producing CXCR2 ligands.	[115]

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
