# Peer review of "The CXCL1-CXCR2 Axis as a Component of Therapy Resistance, a Source of Side Effects in Cancer Treatment, and a Therapeutic Target"

_cancers, 2025, doi:10.3390/cancers17101674_

Round 1

Reviewer 1 Report

Comments and Suggestions for Authors

  • The abstract should be revised. It lacks results. The authors should add some results, preferably numerical results, about the role of CXCL1 in cancer therapy.
  • The graphical abstract is good, but it requires further improvement, such as higher resolution and adding some mechanisms to each cancer-related process.
  • The authors should explain whether the inhibitors presented in Figure 2 are FDA-approved or not.
  • The authors should draw a table or figure to present various available strategies (in research or market) for inhibiting CXCL1 (chemotherapy, immunotherapy, antisense RNA…)
  • I believe the authors should use numbers for sections and subsections.
  • Is there any combination therapy against CXCL1? The authors did not discuss this at all.
  • There is no conclusion in the manuscript.

Author Response

Rev1.

  • The abstract should be revised. It lacks results. The authors should add some results, preferably numerical results, about the role of CXCL1 in cancer therapy.

The abstract has been supplemented with information on the current state of knowledge regarding the described inhibitors in clinical trials.

  • The graphical abstract is good, but it requires further improvement, such as higher resolution and adding some mechanisms to each cancer-related process.

The graphical abstract has been updated to include molecular mechanisms, in accordance with the reviewer’s recommendation.

 The authors should explain whether the inhibitors presented in Figure 2 are FDA-approved or not.

The information has been added.

  • The authors should draw a table or figure to present various available strategies (in research or market) for inhibiting CXCL1 (chemotherapy, immunotherapy, antisense RNA…)

In accordance with the reviewer’s recommendation, a table has been added. Since two sections describe strategies targeting CXCL1–CXCR2, we prepared two separate tables.

  • I believe the authors should use numbers for sections and subsections.

The sections have been numbered in accordance with the reviewer’s recommendation.

  • Is there any combination therapy against CXCL1? The authors did not discuss this at all.

The use of a compound targeting either CXCL1 or CXCR2 in combination with standard anticancer drugs has been described based on the results of clinical studies.

  • There is no conclusion in the manuscript.

The Conclusions section has been added.

Reviewer 2 Report

Comments and Suggestions for Authors

The review article “The significance of CXCL1 in cancer therapy” by Korbecki et al. provides an analysis of recent advancements in cytokine-directed therapy, a rapidly evolving area of cancer therapy, with the focus on CXCL1-CXCR2 axis. Overall, the review accumulates analysis of a wealth of preclinical data and highlights translational potential but would benefit from improved structure and visualization and deeper critical analysis. I recommend a major revision before the publication.

Here are the major points to be addressed:

1) The title of the manuscript seems to be too generous. It can be corrected to make it more accurate.

2) The main drawback of the manuscript is the absence of an overview of the very recent studies and data (2023 - 2025). In case of such an emerging field, it is extremely important to cite recent advancements in CXCL1/CXCR2-focused immune-oncology and small molecule development.

3) Mechanistic and pharmacologic details are described only in text which makes it difficult to track multiple agents, models, and pathways. I recommend adding some tables to improve visualization of the content. For example, it would be beneficial to list key CXCR2 antagonists (SB225002, SCH 527123, SCH 479833, Ladarixin), their mechanism, cancer models tested, and development status. Also, I recommend summarizing CXCL1-mediated resistance mechanisms (for example, upregulation by taxanes via NF-κB, induction of anti-apoptotic Bcl 2 proteins) and corresponding inhibitors.

4) Most data are preclinical. Clinical translation challenges are not comprehensively described. I propose to review any ongoing/completed clinical trials of CXCR2 antagonists (e.g., AZD5069 in inflammatory diseases) and discuss applications in oncology. Also, authors should address potential compensatory upregulation of other ELR+ CXC chemokines and strategies to mitigate redundancy.

5) Section on blood CXCL1 as a diagnostic tool can be expanded with other human cohort studies supported by discussion of sensitivity/specificity for toxicity or response prediction.

6) The manuscript ends abruptly after side effects without any conclusions or outlook provided. Authors should summarize conclusions and future directions in a closing section, critically evaluating gaps (for example, limited presence of clinical trial data for CXCR2 inhibitors) and outlining priority areas (biomarker development, safety profiling, combination strategies).

Considering the incredible relevance of the review topic, I recommend the article for publication after addressing the above-mentioned points.

Author Response

Rev2.

The review article “The significance of CXCL1 in cancer therapy” by Korbecki et al. provides an analysis of recent advancements in cytokine-directed therapy, a rapidly evolving area of cancer therapy, with the focus on CXCL1-CXCR2 axis. Overall, the review accumulates analysis of a wealth of preclinical data and highlights translational potential but would benefit from improved structure and visualization and deeper critical analysis. I recommend a major revision before the publication.

Here are the major points to be addressed:

1) The title of the manuscript seems to be too generous. It can be corrected to make it more accurate.

The title has been revised in accordance with the reviewer’s recommendation.

2) The main drawback of the manuscript is the absence of an overview of the very recent studies and data (2023 - 2025). In case of such an emerging field, it is extremely important to cite recent advancements in CXCL1/CXCR2-focused immune-oncology and small molecule development.

The manuscript has been supplemented with the latest experimental studies on the topic discussed.

3) Mechanistic and pharmacologic details are described only in text which makes it difficult to track multiple agents, models, and pathways. I recommend adding some tables to improve visualization of the content. For example, it would be beneficial to list key CXCR2 antagonists (SB225002, SCH 527123, SCH 479833, Ladarixin), their mechanism, cancer models tested, and development status. Also, I recommend summarizing CXCL1-mediated resistance mechanisms (for example, upregulation by taxanes via NF-κB, induction of anti-apoptotic Bcl 2 proteins) and corresponding inhibitors.

A table describing CXCR2 inhibitors and compounds targeting CXCL1 has been added in accordance with the reviewer’s recommendation. Due to the close interconnections between all mechanisms listed in the section on chemoresistance, the figure has been expanded to include additional mechanisms.

4) Most data are preclinical. Clinical translation challenges are not comprehensively described. I propose to review any ongoing/completed clinical trials of CXCR2 antagonists (e.g., AZD5069 in inflammatory diseases) and discuss applications in oncology. Also, authors should address potential compensatory upregulation of other ELR+ CXC chemokines and strategies to mitigate redundancy.

Greater emphasis has been placed on clinical studies. For this reason, the section describing available compounds targeting CXCR2 and CXCL1 has been expanded.

5) Section on blood CXCL1 as a diagnostic tool can be expanded with other human cohort studies supported by discussion of sensitivity/specificity for toxicity or response prediction.

No detailed studies have been conducted on the sensitivity or specificity of the association between blood CXCL1 levels and patient response to treatment or treatment-related side effects. Therefore, we are unable to include such a section.

6) The manuscript ends abruptly after side effects without any conclusions or outlook provided. Authors should summarize conclusions and future directions in a closing section, critically evaluating gaps (for example, limited presence of clinical trial data for CXCR2 inhibitors) and outlining priority areas (biomarker development, safety profiling, combination strategies).

The Conclusions section has been added.

Considering the incredible relevance of the review topic, I recommend the article for publication after addressing the above-mentioned points.

Round 2

Reviewer 1 Report

Comments and Suggestions for Authors

The authors made the corrections well, and I recommend publishing them. 

Reviewer 2 Report

Comments and Suggestions for Authors

I appreciate the Authors for providing accurate and comprehensive responses. The manuscript has been significantly improved and can be approved for publication in the present form.